# An Angiogenesis-Related lncRNA Signature Is Associated with Prognosis and Tumor Immune Microenvironment in Breast Cancer

**DOI:** 10.3390/jpm13030513

**Published:** 2023-03-13

**Authors:** Shun Gao, Yuan Wang, Yingkun Xu, Shengchun Liu

**Affiliations:** Department of Breast and Thyroid Surgery, The First Affiliated Hospital of Chongqing Medical University, Chongqing 400016, China

**Keywords:** breast cancer, angiogenesis, lncRNA, tumor immune microenvironment, prognosis

## Abstract

Angiogenesis is crucial in the development and progression of tumors. This study examined the relationship between angiogenesis-related lncRNAs (AR-lncRNAs) and breast cancer (BC) immunity and prognosis. We used univariate Cox regression analysis to obtain AR-lncRNAs closely related to BC prognosis. Cluster analysis of BC patients was performed using non-negative matrix factorization (NMF) analysis according to the expression of AR-lncRNAs that were prognostically relevant. An AR-lncRNA risk model (AR-lncM) was created using LASSO regression analysis to predict the prognosis and survival of BC patients. Subsequently, the effect of LINC01614 on cell migration and invasion was verified by Transwell and Western blot assays, and the CCK-8 assay detected its impact on cell sensitivity to tamoxifen. Finally, we obtained 17 AR-lncRNAs from the TCGA database that were closely associated with the prognosis of BC patients. Based on the expression of these AR-lncRNAs, BC patients were divided into five clusters using NMF analysis. Cluster 1 was found to have a better prognosis, higher expression of immune checkpoints, and higher levels of immune cell infiltration. Furthermore, an AR-LncM model was created using ten prognostic-related AR-lncRNAs. The model’s risk predictive performance was validated using survival analysis, timeROC curves, and univariate and multivariate Cox analysis. The most interesting gene in the model, LINC01614, was found to regulate epithelial-mesenchymal transition (EMT) and tamoxifen sensitivity in BC cells, implying that LINC01614 could be a potential therapeutic target for BC patients.

## 1. Introduction

Globally, BC (breast cancer) remains the most common tumor (30%) and the leading cause of cancer-related death (15%) in female patients, despite strategies for early detection, early prevention, and early treatment [1]. Furthermore, as a heterogeneous tumor, patients with BC with the same clinical characteristics may have completely different prognoses [2]. Therefore, it is necessary to identify new prognostic factors and develop more accurate prognostic models to help optimize individualized treatment.

Angiogenesis is the process by which existing capillaries or post-capillary veins develop to form new blood vessels [3]. In this process, pro-angiogenic and anti-angiogenic factors are essential and are called “angiogenic switches” [4]. In a healthy state, the “switch” is silent. However, tumor cells can induce angiogenesis in vivo [5,6]. Once the “switch” is activated, new blood vessels deliver oxygen and nutrients to tumor cells, promoting tumor growth, metastasis, and invasion [6,7]. Therefore, angiogenesis significantly impacts the occurrence and development of cancer [8]. In recent years, it has been reported that signatures constructed using angiogenesis-related genes can successfully predict survival and prognosis of tumor patients, for example, Chen et al. [9]. constructed a prognostic signature composed of four angiogenesis-related long noncoding RNAs (AR-lncRNAs) related to the survival of gastric cancer patients, showing the high efficiency of angiogenesis-related prognostic signatures. Similarly, Lei et al. [10]. constructed a five-AR-lncRNA model, which can be used to predict the prognoses of patients with liver cancer. Indeed, the importance of angiogenesis in BC progression has been demonstrated. Inhibiting angiogenesis in BC, promoting the normalization of tumor blood vessels, and re-editing the tumor microenvironment (TME) are effective ways to treat BC [11,12,13,14]. For example, bevacizumab, which antagonizes VEGF, and lapatinib, which inhibits VEGFR2, have been used to treat triple-negative BC [15], but the therapeutic effect is still limited. Tumor immune responses are closely related to angiogenesis, and tumor angiogenesis is highly dependent on the immunosuppressive microenvironment. The presence of tumor-infiltrating lymphocytes (TILs) in the tumor microenvironment is a marker of effective tumor cure. According to the state of TIL, the tumor microenvironment can be divided into three types: (1) immune-inflammatory type: functional CD8+ T densely infiltrated cells; (2) non-infiltrating type: abnormal angiogenesis and immunosuppressive microenvironment to prevent the infiltration of T cells; and (3) immunocompromised type [16]. Tumors of the immune-inflammatory type respond better to immune checkpoint inhibitors (ICIs) than other types [17]. Therefore, improving T-cell infiltration in tumor tissue can improve ICI efficacy.

In this study, we analyzed AR-lncRNA expression in BC and constructed a prognostic model of AR-lncRNA. The roles of immune cell infiltration and microenvironmental heterogeneity in different BC clusters and risk subgroups were explored. In addition, the biological function of LINC01614 was tested in vitro. We discovered that LINC01614 could regulate the migration and invasion of BC cells by regulating epithelial-mesenchymal transition (EMT) and the sensitivity of BC cells to tamoxifen.

## 2. Materials and Methods

### 2.1. Acquisition and Arrangement of Raw Data

RNA-seq expression and clinical information of breast cancer patients were obtained from the TCGA database. The 1078 BC samples (Appendix A) from the TCGA database were then randomly assigned to the training set (n = 539) and validation set (n = 539) using the R language for the corresponding study. A total of 104 angiogenesis-related genes were obtained from the molecular signature database (https://broadinstitute.org/gsea/msigdb/, accessed on 20 May 2022) (details in Table 1).

### 2.2. Identification of AR-lncRNAs 

Spearman correlation analysis was performed to assess the correlation of angiogenesis-related lncRNAs with angiogenesis-related genes, and AR-lncRNAs were identified according to the criteria of correlation coefficients with absolute values > 0.3 and *p*-values < 0.001 (|R| > 0.3, *p* < 0.001). To explore the prognostic value of AR-lncRNAs, univariate Cox regression analysis was performed on AR-lncRNAs associated with patients’ overall survival (OS), and HR values and Cox *p*-values were calculated.

### 2.3. Definition of BC Subtypes by AR-lncRNAs

Clustering was performed using non-negative matrix factorization (NMF) analysis, BC patients were classified into different clusters, and differences between clusters were then analyzed. Differences in clinical characteristics and immune checkpoint membership between different clusters were assessed using chi-square tests.

### 2.4. Assessment of Immune Infiltration

The CIBERSORT algorithm was used to obtain the infiltration levels of different immune cells. Gene expression data were uploaded to the CIBERSORT website (http://cibersort.stanford.edu/, accessed on 22 May 2022), and the processed results (*p* < 0.05) were used for analysis. The ESTIMATE algorithm was used to quantify the immune and stromal components of tumors. In this algorithm, immune and stromal scores are calculated by analyzing specific gene expression profiles of immune and stromal cells, and the ESTIMATE score is defined as the combination (i.e., sum) of immune and stromal scores and can be thought of as a “non-tumor score”. Consequently, a high ESTIMATE enrichment gives a low tumor purity score and vice versa [18].

### 2.5. Analysis of Gene Set Enrichment

To identify LINC016014-associated biological processes and pathways, GSEA was performed using the TCGA overall dataset; all patients were divided into high- and low- LINC016014-expression groups according to median values, and the analysis was performed using the GSEA 4.1.0 tool [19]. The gene set for the canonical pathway (c2.cp.kegg.v7.2.symbols.gmt) was obtained from the Molecular Signature Database (http://software.broadinstitute.org/gsea/msigdb/index.jsp accessed 24 May 2022). Gene set permutations were performed 1000 times for each analysis to obtain a normalized enrichment score (NES), which was used for sorting pathways enriched in each phenotype. A result was regarded as significant when the nominal *p*-value was <0.05 and the false discovery rate (FDR) was <0.2. A similar method was used in the analysis of biological processes and pathways between different clusters, except that the two sides of the comparison were replaced by cluster 1 and the other 4 clusters.

### 2.6. Evaluation and Verification of the Accuracy of Prognostic Signatures

Based on the 17 prognosis-related AR-lncRNAs obtained from the TCGA dataset, the best AR-lncRNA model was established by LASSO Cox analysis [20]. The formula used was as follows: risk score = (lncRNA1 coefficient × lncRNA1 expression) + (lncRNA2 coefficient × lncRNA2 expression) + … + (lncRNAn coefficient × lncRNAn expression). According to the model, the training set was divided into high- (270) and low-risk (269) groups (Appendix A). The survival differences between the two groups were determined using Kaplan–Meier survival analysis. The timeROC curve [21,22] and Cox regression analysis were used to assess the model’s predictive performance.

### 2.7. The Creation of the Nomogram

A nomogram was created based on patients’ clinical characteristics (included the patient’s lymph node involvement, the size of the patient’s tumor, and the patient’s age) and risk scores. Among all BC patients, there were few patients with distant tumor metastasis, so patient tumor metastasis status was not applicable in the construction of the nomogram. The principle of drawing a nomogram is to draw a horizontal line, determine its position according to the characteristics of each variable, add the points of all variables to obtain a total score, and normalize it to a distribution of 0 to 100.

### 2.8. Prediction of Response to Immune Checkpoint Inhibition (ICI) Therapy

The Tumor Immune Dysfunction and Exclusion (TIDE) score, an assessment scheme that can be used to predict the potential therapeutic responsiveness of oncology patients to ICI, is a computational algorithm based on gene expression profiling (http://tide.dfci.harvard.edu, accessed on 28 May 2022). Patients with higher TIDE scores are less sensitive to ICI drugs, thus indicating a high probability of antitumor immune escape [23]. The details of the algorithm are described in detail in other studies in the literature [23].

### 2.9. Patients and Tissue Specimens

The First Affiliated Hospital of Chongqing Medical University provided 12 pairs of human breast tissue specimens (details in Table 2). All patients provided their written informed consent. The Ethics Committee of Chongqing Medical University approved the study (approval number: K2023-058).

### 2.10. Cell Culture and Treatment

BC cell lines (MCF-7, BT-549, MDA-MB-231, and T47D) and HUVECs were grown in 10% FBS-containing Dulbecco’s Modified Eagle’s Medium (DMEM) (Gibco, MA, USA). All cells were cultured and transfected as directed by the manufacturer. 

The interfering antisense oligonucleotide probes (ASO probes) (GenePharma, Shanghai, China) with LINC01614 were as follows:

ASO-LINC01614#1: ATTACGAAATGCTTCCAGC;

ASO-LINC01614#2: TTTAATAGGAGAAACCCTC;

ASO-LINC01614#3: ATAAATCACAGAACCAGCC;

ASO-NC: GCGUATTATATTATAGCCGATTAAC.

### 2.11. Quantitative Real-Time PCR

Total RNA was extracted using an RNA extraction kit (FOREGENE, Chengdu, China), and cDNA samples were synthesized using a reverse transcription kit (Takara Biotechnology Co., Ltd., Beijing, China). Real-time PCR was used to measure the expression of LINC01614. β-actin was identified as an internal control.

The LINC01614 primers used were as follows:

Forward: 5′-GAGCAGAATCACCACCTCACA-3′;

Reverse: 5′-TAGCAGGTGAAGGCACCCTA-3′.

The β-actin primers used were as follows:

Forward: 5′-CCTTCCTGGGCATGGAGTC-3′;

Reverse: 5′-TGATCTTCATTGTGCTGGGTG-3′.

### 2.12. Tamoxifen-Sensitization Assays

In our study, we used the “pRRophetic” R package to calculate the IC50 of each sample and validated the tamoxifen sensitivity on the MCF-7 cell line. We first transiently transfected cells (NC, ASO-LINC01614#1, ASO-LINC01614#2) and subsequently treated the indicated concentrations of tamoxifen (Sigma-Aldrich, MO, USA) (0 μM, 5 μM, 10 μM, 15 μM, 20 μM, and 25 μM). The cells were cultured for 48 h, and, finally, the change in drug sensitivity was detected using a CCK-8 kit (CCK-8; Dojindo, Kumamoto, Japan).

### 2.13. Tube Formation Assay

Melted Matrigel (BD Biosciences, Franklin Lakes, NJ, USA) was evenly distributed in a 96-well plate (50 L/well) and solidified at 37 °C for 1 h. HUVECs were homogeneously mixed in DMEM and seeded into pretreated 96-well plates (1 × 10^5^ cells/well). After 6 h in culture, pictures of the cells were taken.

### 2.14. Assays for Cell Migration and Invasion

Transwell assays were used to test cell migration abilities. After 48 h of cell transfection (NC, ASO-LINC01614#1, ASO-LINC01614#2), cells were harvested and resuspended in DMEM before being cultured for another 24 h. The chambers (Corning Incorporated, Corning, NY, USA) were taken out after the time point was reached. Cells were fixed with paraformaldehyde, stained with crystal violet, and photographed under a microscope to produce pictures. The same method was used to detect changes in cell invasiveness, except for the addition of a Matrigel layer to the chambers’ upper layer.

### 2.15. Immunoblots

Total proteins were extracted from breast cancer cells and then analyzed using Western blotting, as directed by the manufacturer. The antibodies used were E-cadherin, N-cadherin, and vimentin (cat. nos. 3195T, 13116T, and 5741T; dilution 1:1000; Cell Signaling Technology, Danvers, MA, USA). GAPDH (cat. no. AB0037, 1:5000; Abways, Shanghai, China) was used as an internal control.

### 2.16. Statistical Analysis

The continuous variables in the study were displayed as means and standard deviations. For statistical analysis, GraphPad Prism 9 and R software were used. For statistical comparisons, the analysis of variance or *t*-tests were used. * *p* < 0.05 was regarded as statistically significant.

## 3. Results

### 3.1. Selection of AR-lncRNAs in the Cancer Genome Atlas BC Cohort

RNA-Seq data were obtained from the TCGA-BRCA dataset, and the expression of 104 angiogenesis-related genes in breast cancer tissues and adjacent tissues were plotted using a heatmap (Figure 1A). Subsequently, an interaction network map among these genes was generated using the STRING database (Figure 1B,C). Next, 699 lncRNAs significantly associated with angiogenesis-related genes were identified using Spearman correlation analysis and visualized using Cytoscape (Figure 1D). Furthermore, the clinical significance of the AR-lncRNAs was investigated using univariate Cox regression analysis, and the results showed that 17 AR-lncRNAs were strongly associated with OS in BC patients (Figure 1E). Furthermore, the heatmap (Figure 1F) and boxplot (Figure 1G) showed the differential expression of 17 AR-lncRNAs in TCGA BC tissues and normal tissues.

### 3.2. Identification of AR-lncRNAs and Immune Subtypes of BC

Molecular typing has developed into an effective strategy to provide patients with the best treatment options and is widely used in clinical diagnosis and treatment today. Therefore, we performed NMF analysis based on the expression information of 17 AR-lncRNAs associated with prognosis. After analyzing the results, we chose k = 5 to group BC patients (Figure 2A). The patients’ combined OS and progression-free survival (PFS) analysis indicated that cluster 1 had the best prognosis and that cluster 3 had the worst (Figure 2B,C). Subsequently, the relationship between clinical characteristics and BC clusters was investigated (Figure 2D). Furthermore, we explored the differential expression of immune checkpoints in different BC clusters. Cluster 1 patients had significantly higher expressions of PD-L1, PD-1, CTLA-4, Tim3, LAG3, ICOS, and IDO1 than the other four clusters (Figure 2E–K). We then investigated the relationship between PD-L1 and 17 AR-lncRNAs. PD-L1 was significantly associated with six AR-lncRNAs (Linc01871, Linc02166, CDK6-AS1, USP30-AS1, PRKCZ-AS1, and LMNTD2-AS1) (Figure 2L). In addition, the correlations between other immune checkpoints and 17 AR-lncRNAs are presented in Appendix A. Taken together, these results suggest that the BC patients in cluster 1 had higher immunogenicity, suggesting that they would be more likely to respond to ICI therapy.

### 3.3. Immune Landscape of AR-lncRNA Subtypes of BC

The CIBERSORT algorithm assessed the degree of immune infiltration in different BC clusters. The results indicated that cluster 1 had higher immune, stromal, and ESTIMATE scores but lower tumor purity (Figure 3A–D). Individual levels of immune infiltration in different BC clusters were assessed using ESTIMATE. Cluster 1 had more activated CD4 memory T cells and CD8 T cell infiltration than other BC clusters (Figure 3E,F). In addition, cluster 1 had more follicular helper T cells and regulatory T cell infiltration relative to clusters 2, 4, and 5 (Figure 3G,H). Likewise, we observed a low infiltration of resting mast cells, M0 macrophages, and M2 macrophages and a high infiltration of M1 macrophages in cluster 1 (Figure 3I–L). The above results again suggest that cluster 1 is a subtype with a stronger ability for immune response.

### 3.4. Analysis of Gene Set Enrichment

GSEA was used to explore the differences in biological processes between cluster 1 and the other clusters. The results suggested that various immune processes were highly enriched in cluster 1 compared to the other groups. The GO signatures revealed that humoral immune responses, regulatory signaling pathways for the immune response, natural killer cell activation, T cell activation involved in immune responses, and production of molecular mediators for the immune response were positively correlated in cluster 1 (Figure 4A–M). The KEGG signature also showed that the cytotoxicity mediated by B cell natural killer cells, the receptor signaling pathway, and the T cell receptor signaling pathway were positively correlated in cluster 1 (Figure 4N–P).

### 3.5. Generation of an AR-lncRNA Model for Prognosis

We performed a LASSO regression analysis on 17 prognostic AR-lncRNAs in the TCGA BRCA dataset and constructed a model of angiogenesis-associated lncRNAs (AR-LncM) linked to OS in BC patients. LASSO regression analysis identified ten lncRNAs as AR-lncRNAs closely related to prognosis (Figure 5A,B). Among these, seven AR-lncRNAs (Linc02166, Linc01615, WARS2-IT1, CDK6-AS1, NIFK-AS1, NDUFA6—DT, and LMNTD2-AS1) were closely related to AR-LncM risk scores (Figure 5C). Nomograms were created based on the patient’s clinical features and AR-LncM risk scores (Figure 5D). The model’s prediction accuracy was evaluated using calibration curves (Figure 5E), and timeROC curve analysis produced an AUC value of 0.785 for the AR-LncM risk score (Figure 5F). 

Additionally, we explored BC patients after randomly assigning them to training set (N = 539) and validation set (N = 539). Patients were divided into high- and low-risk groups based on the median risk scores determined for each group using a coefficient of 10 AR-lncRNAs (details in Table 3). The distribution of risk scores and OS statuses for the entire TCGA set, the training set, and the validation set are shown in Figure 5G–I. Further evidence from the timeROC curves demonstrated that the TCGA’s AR-LncM had a high predictive performance on the entire training and validation sets (Figure 5J–L). Patients in the high-risk group had shorter OS in the whole TCGA set and the training and validation sets, according to the survival analysis (Figure 5M–O). These findings suggest that AR-LncM is prognostic for the survival of BC patients.

### 3.6. The AR-LncM Signature’s Predictive Value

The patient’s clinical features and AR-LncM risk scores were evaluated using univariate and multivariate Cox regression score risk analyses. The results showed that the AR-LncM scores were significantly correlated with OS in BC patients (Figure 6A–D), suggesting that the effect of AR-LncM on patient survival prognosis may function as a separate risk factor. Furthermore, a stratified analysis of the clinical characteristics of BC patients showed that, for all subgroups, the high-risk group had a shorter OS than the low-risk group (Figure 6E–M).

### 3.7. Analysis of the Function and Clinical Characteristics of AR-LncM

To explore the function and clinical characteristics of AR-LncM, the expression of AR-lncRNAs in cancer tissues of breast cancer patients was visualized using Sankey plots (Figure 7A). Gene enrichment analysis showed that drug metabolism cytochrome, ECM receptor interaction, focal adhesion, PPAR signaling pathway, and steroid hormone biosynthesis were enriched in the group with high-risk scores (Figure 7B). In contrast, allograft rejection, autoimmune thyroid disease, graft versus host disease, hematopoietic cell lineage, and primary immunodeficiency were enriched in the group with low-risk scores (Figure 7C). Additionally, we explored the correlation between clinical characteristics and AR-LncM signatures, and the results showed that patients in cluster 1 (Figure 7D), patients with high immune scores (Figure 7E), and patients with stage I–II (Figure 7F) had low risk scores. However, the risk scores did not differ in assessing distant metastasis, lymph node involvement, tumor size, or age in breast cancer patients (Figure 7G–J). Similarly, we also observed the same results in the heatmap (Figure 7K).

### 3.8. Relationship between the AR-LncM Signature and Immunotherapy and Immune Cell Infiltration

The association between AR-lncM and immune cell infiltration and immunotherapeutic response was investigated. First, we examined the association between checkpoint expression and high- and low-risk groups, and we found that the expressions of PD-L1, PD-1, CTLA4, LAG3, IDO1, and ICOS were increased in the low-risk group, as shown in Figure 8A–F. Next, CIBERSORT was used to compare immune cell infiltration levels in the two risk groups. The results suggested that the high-risk group had higher levels of M0 and M2 macrophages but lower levels of CD8 T cells, follicular helper T cells, regulatory T cells, resting NK cells, and activated NK cells (Figure 8G). 

Patients with higher infiltration of M0 and M2 macrophages had a worse prognosis (Figure 8H–I). In contrast, patients with higher infiltration of naive B cells, plasma cells, and resting memory CD4 T cells had a better prognosis (Figure 8J–L). Next, the correlation analysis of the ssGSEA training cohort showed that, in the low-risk group, immune checkpoints and T helper cells were significantly activated, while only macrophages were significantly inhibited (Figure 8M). On the other hand, the TIDE algorithm was used to assess the correlation between risk groups and potential effectiveness of ICI. The results showed that the low-risk group had higher TIDE scores (Figure 8N), lower TIDE exclusion scores (Figure 8O), and higher TIDE dysfunction scores (Figure 8P), suggesting a greater possibility of tumor immune escape in the low-risk group. These findings imply that ICI therapy may be more effective in high-risk patients.

### 3.9. LINC01614 Is Involved in Regulating Cell Invasion, Migration, and Drug Sensitivity 

Although LINC01614, LINC01615, LINC02166, and LMNTD2-AS1 were all overexpressed in BC tissues (Appendix A), only overexpression of LINC01614 predicted a poor prognosis for BC patients (Figure 9A,B). Similarly, the expression of LINC01614 in cancerous and adjacent tissues of 12 BC patients was examined by PCR assay, which confirmed that LINC01614 was highly expressed in BC (Figure 9C). Furthermore, using GSEA, we discovered that LINC01614 was associated with angiogenesis, EMT, and the TGF-signaling pathway (Figure 9D–G). This indicates that LINC001614 may be a significant risk factor for BC patients. Next, we found that, in BC cell lines, LINC01614 was expressed most highly in MCF-7 cells (Figure 9H), so we knocked down LINC01614 in MCF-7 with antisense oligonucleotides (ASOs) and performed PCR to validate this (Figure 9I). Next, using a tube formation experiment to mimic the BC tumor microenvironment, we co-cultured MCF-7 and HUVEC cells, and found that LINC01614 was undoubtedly involved in angiogenesis (Figure 9J).

The Transwell assay found that the knockdown of LINC01614 showed a significant reduction in the migratory and invasive abilities of MCF-7 compared to the those of the control group (Figure 9K–M). Similarly, Western blot analysis revealed that LINC01614 knockdown increased E-cadherin expression while decreasing the expression of N-cadherin and Vimentin (Figure 9N), confirming that LINC01614 regulates EMT. Tamoxifen resistance is the leading cause of a poor prognosis in some ER-positive BC patients. It also presents difficulties in clinical diagnosis and treatment. Therefore, we explored whether LINC01614 was involved in regulating tamoxifen resistance. Bioinformatics analysis suggested that patients with higher LINC01614 expression were significantly less sensitive to tamoxifen (Figure 9O), which was confirmed by the CCK-8 assay in MCF-7 cells (Figure 9P). These results suggest that LINC01614 can affect not only the migratory ability of MCF-7 cells by regulating EMT but also the sensitivity of MCF-7 cells to tamoxifen, implying that LINC01614 may be a potential target for BC patients.

## 4. Discussion

Angiogenesis is a complex, multistep biological process in which new capillaries grow from pre-existing blood vessels to supply tissues with oxygen and nutrients [5]. Angiogenesis in tumors mediates tumor invasion into the adjacent stroma, promoting tumor metastasis. In most tumors, microvascular formation is a prognostic marker for metastasis, recurrence, and survival [24,25]. Angiogenesis-related genes have been reported to be potential prognostic markers associated with BC, and integrating these genes has important implications for understanding and assessing tumor progression [1,26,27,28,29,30]. Among these genes, lncRNAs also play an essential role [26,28,31]. However, the potential role of AR-lncRNA signatures as an effective therapeutic strategy for BC remains poorly understood. Therefore, we attempted to create an AR-lncRNA signature prediction model to generate new ideas for the individualized treatment of BC.

The predictive value of AR-lncRNAs in BC was investigated in this study. Expression data for 104 angiogenesis-related genes and 699 lncRNAs from the TCGA-BRCA cohort were extracted. Seventeen AR-lncRNAs associated with prognosis were obtained using Spearman’s correlation analysis and univariate Cox regression analysis. Furthermore, BC patients were divided into five clusters by NMF analysis, and it was observed that immune checkpoints showed high expression in cluster 1. CIBERSORT and ESTIMATE analyses revealed that cluster 1 had a higher level of immune cell infiltration. GSEA revealed that various immune processes were significantly over-represented in cluster 1. Subsequently, 10 AR-lncRNAs were chosen by LASSO regression analysis to establish AR-LncM, and survival analysis and timeROC curves confirmed AR-LncM’s good prognostic value and predictive performance.

AR-LncM was also found to be an independent risk factor in univariate and multivariate Cox regression analyses. Next, based on variations in patient risk scores, BC patients were split into high- and low-risk groups. The two risk groups displayed various clinical characteristics and levels of immune cell infiltration. The TIDE algorithm was used to assess the potential clinical response of BC patients to ICI therapy. The results revealed that ICI therapy might be effective for high-risk BC patients. We discovered a highly immunogenic BC cluster and investigated a novel AR-LncM signature that could be used as a biomarker for BC-based AR-lncRNAs. To analyze the potential mechanisms of lncRNAs involved in angiogenesis and adverse progression, we performed gene enrichment scoring of the most interesting lncRNA in the model. We showed that LINC01614 was closely associated with the EMT and TGF-β signaling pathways. Transwell assays confirmed that the knockdown of LINC01614 resulted in MCF-7 cells with significantly reduced migration and invasion abilities. The Western blot assay showed that this could have been caused by LINC01614 regulating EMT. Furthermore, the CCK-8 assay revealed that LINC01614 knockdown increased MCF-7 cell sensitivity to tamoxifen.

The construction of risk models for angiogenesis-related genes has been reported in BC; for example, Tao et al. [30]. constructed a risk model consisting of four angiogenesis genes (TNFSF12, SCG2, COL4A3, and TNNI3) which showed good predictive performance in the prognosis of BC patients. In addition, Xu et al. [32] constructed a risk model that included seven genes (BTG1, IL18, PF4, RUNX1, SCG2, THY1, and TNFSF12), which also showed good accuracy in predicting the survival of BC patients. The results of the above two groups of investigators suggest that the construction of risk models for angiogenesis-related genes in BC patients can provide new guidance on therapeutic targets for patients. Our study focused on the study of angiogenesis-related lncRNAs in BC, and our AR-lncM was found to independently predict OS and immunotherapeutic effects in the TCGA-BRCA dataset. An interesting gene from the model, LINC01614, was validated in BC cells, and LINC01614 was observed to have the BC target characteristics.

The TME is representative of a complex and dynamic environment of cellular and cell-free components with synergistic responses and functions in cancer progression, which are closely related to tumor treatment [33,34,35]. Various immune cells can function as tumor suppressors or promoters during BC progression [1,36,37,38]. It has been reported that angiogenesis is involved in the interaction of products in the TIME; for example, tumor-associated macrophages (TAMs) have been reported to mediate angiogenesis by secreting growth and inflammatory factors [39]. However, in our study, we found that M2 macrophages (defined as TAMs) had a higher level of infiltration in high-risk BC patients, consistent with previous reports that M2 macrophages can promote BC progression [40,41]. Furthermore, NK cells and DC cells have also been reported to directly or indirectly regulate angiogenesis [42]. However, in the present study, NK cells in high-risk BC patients infiltrated less. In contrast, the levels of DC cell infiltration did not exhibit differences between the two risk groups, suggesting an important role for NK cells in BC angiogenic effects. Previous studies have shown that Tregs, CSCs, and CD8/Treg ratios are positively associated with BC disease progression [43]; however, in the present study, Tregs and CD8 T-cell infiltration were lower in the high-risk BC patients, possibly due to the more important role of other immune cells and tumor-promoting regulators in the TIME. The above results suggest that immune cells have an important influence on BC progression, and an in-depth investigation targeting the TIME would be beneficial for the treatment of BC in the clinic. 

Although we observed some interesting phenomena, we acknowledge that this study has significant limitations, such that further investigations are needed to make the conclusions more reliable. For example, with our small sample size and multiple testing, the assessment of the expression of LINC01614 and its impact on BC prognosis may have led to false-positive results. Second, we did not further explore the function of LINC01614 in regulating angiogenesis, because such research still needs to be verified by co-culture of cells or exosomes to treat cells; however, we used transient transfection to treat BC cells, so it is temporarily impossible to carry out such research. We will conduct further validations after establishing a stable LINC01614 knockdown cell line in the follow-up work.

## 5. Conclusions

This study looked into the prognostic value of AR-lncRNAs in BC. A cluster of highly immunogenic BC patients was identified, and a risk model consisting of AR-lncRNAs was constructed. The model independently predicted OS and immunotherapy effects in the TCGA-BRCA dataset, and LINC01614, an interesting gene in the model, was validated in BC cells.

## Figures and Tables

**Figure 1 jpm-13-00513-f001:**
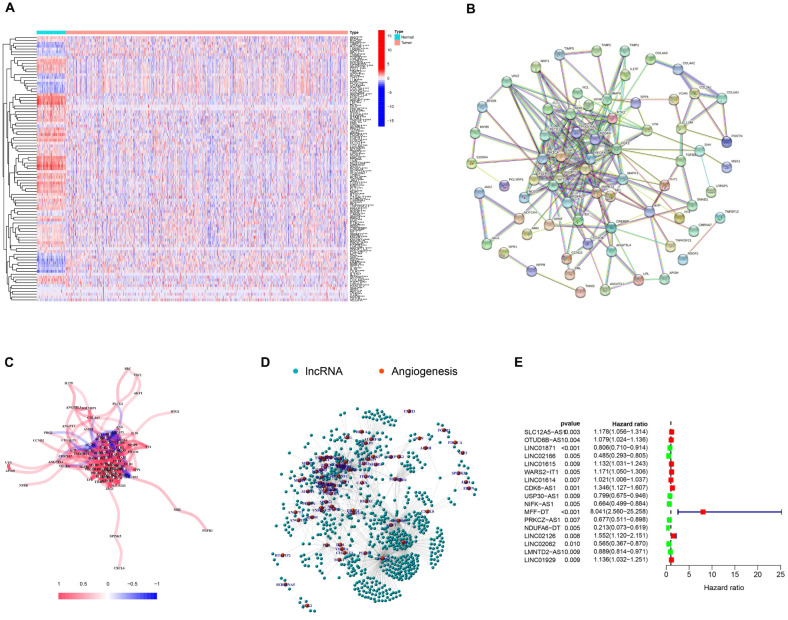
Identification of AR-lncRNAs in BC patients. (**A**) Heatmap of angiogenesis-related genes in BC tissues and normal tissues. (**B**) The interactions among candidate angiogenesis-related genes are shown by the PPI network. (**C**) The correlation network of candidate genes. (**D**) The network of 699 AR-lncRNAs. (**E**) The forest plot of 17 prognostic AR-lncRNAs. (**F**,**G**) The heatmap and box plot of 17 prognostic AR-lncRNAs in both tumor and normal samples. * *p* < 0.05, ** *p* < 0.01, and **** p* < 0.001.

**Figure 2 jpm-13-00513-f002:**
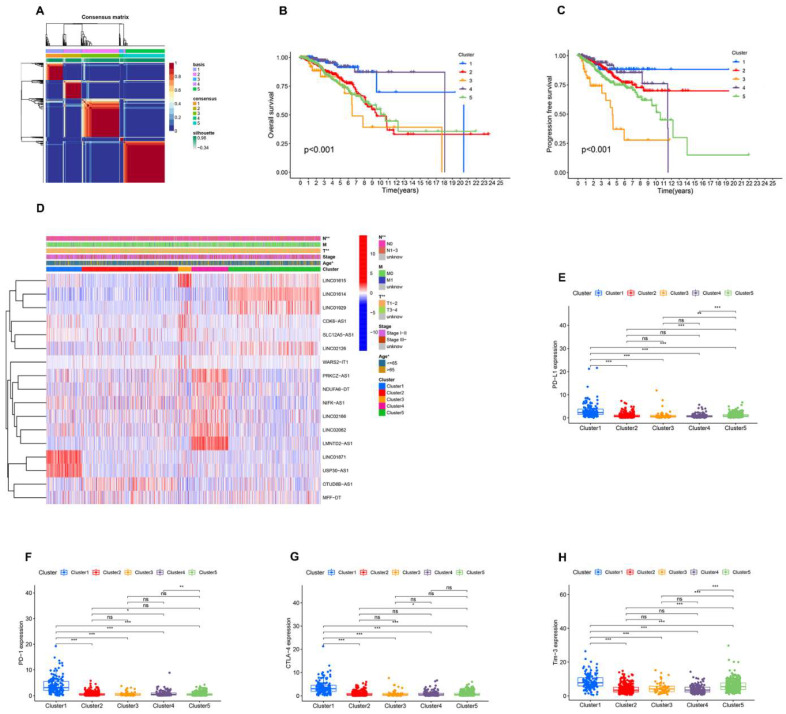
NMF analysis based on prognosis-related AR-lncRNAs in BC. (**A**) Consensus map for NMF clustering. (**B**,**C**) Kaplan–Meier curves of OS and PFS for five clusters. (**D**) Heatmap and clinicopathologic features of the five clusters. (**E**–**K**) Differential expression of immune checkpoints PD-L1, PD-1, CTLA-4, Tim3, LAG3, ICOS, and IDO1 in different clusters. (**L**) Correlation of PD-L1 and 17 prognostic AR-lncRNAs. (Red indicates a positive correlation, blue indicates a negative correlation, and an asterisk indicates statistical significance. The darker the color, the more obvious the significance) ns, no significance; * *p* < 0.05, ** *p* < 0.01, and **** p* < 0.001.

**Figure 3 jpm-13-00513-f003:**
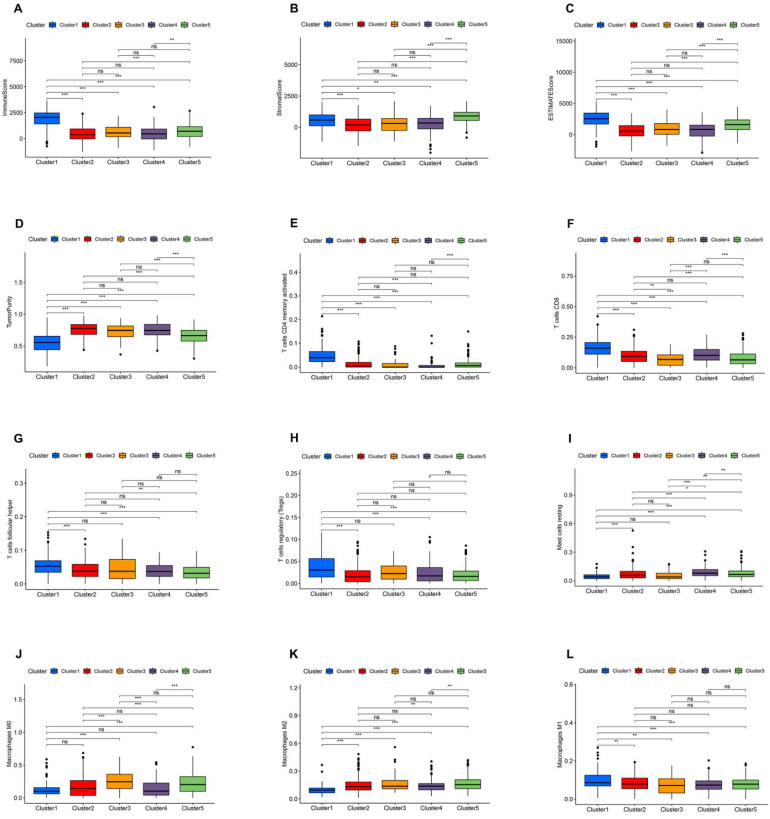
Immune signatures of five AR-lncRNA BC clusters. (**A**–**D**) Different expression of immune score (**A**), immune score (**B**), stromal score (**C**), ESTIMATE score, and Tumor Purity (**D**) in five BC clusters. (**E**–**L**) Differences in the levels of infiltration of CD4 memory T cells, CD8 T cells, follicular helper T cells, regulatory T cells, resting mast cells, M0 macrophages, M2 macrophages, and M1 macrophages in five BC clusters. ns, no significance; * *p* < 0.05, ** *p* < 0.01, and *** *p* < 0.001.

**Figure 4 jpm-13-00513-f004:**
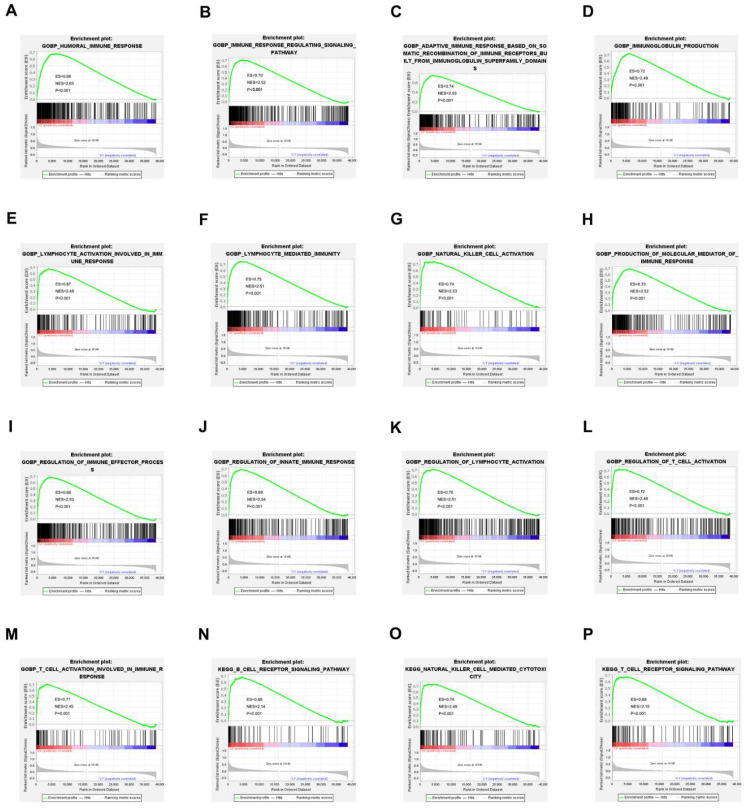
GSEA shows that BC cluster 1 is enriched for various immune-response processes compared to other clusters (**A**–**P**). The results showed that biological processes, such as humoral immune response, the immune response regulating signaling pathway, natural killer cell activation, T cell activation involved in immune response, production of molecular mediator of the immune response, the B cell receptor signaling pathway, natural killer cell-mediated cytotoxicity, and the T cell receptor signaling pathway, and so on, were highly enriched in BC cluster 1.

**Figure 5 jpm-13-00513-f005:**
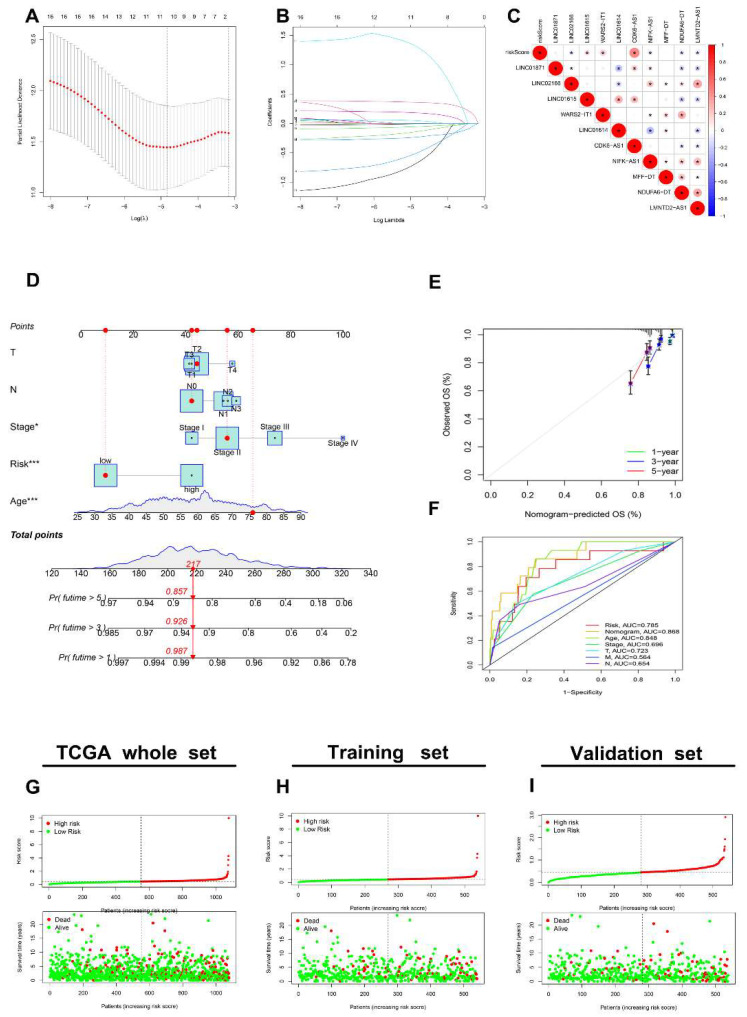
Risk model for AR-lncRNAs. (**A**) LASSO coefficient profiles of 17 prognostic AR-lncRNAs in the TCGA training cohort. (**B**) Through the LASSO regression model, AR-lncRNAs with the best discriminative power (10) were selected for establishing risk scores. (**C**) Correlations between risk scores and AR-lncRNAs in the model were determined using Spearman correlation analysis. (**D**) The nomogram model assesses the 1-, 3-, and 5-year survival probability of BC patients. (**E**) The 1-year, 3-year, and 5-year survival-rate calibration curves of the line chart. (**F**) TimeROC curves of nomograms, risk scores, and clinical characteristics in the TCGA dataset. (**G**–**I**) The BC patients were divided into two groups according to risk scores. (**J**–**L**) The diagnostic efficacy of AR-lncM was assessed by timeROC curves. (**M**–**O**) Differences in OS between high- and low-risk groups. * *p* < 0.05, and **** p* < 0.001.

**Figure 6 jpm-13-00513-f006:**
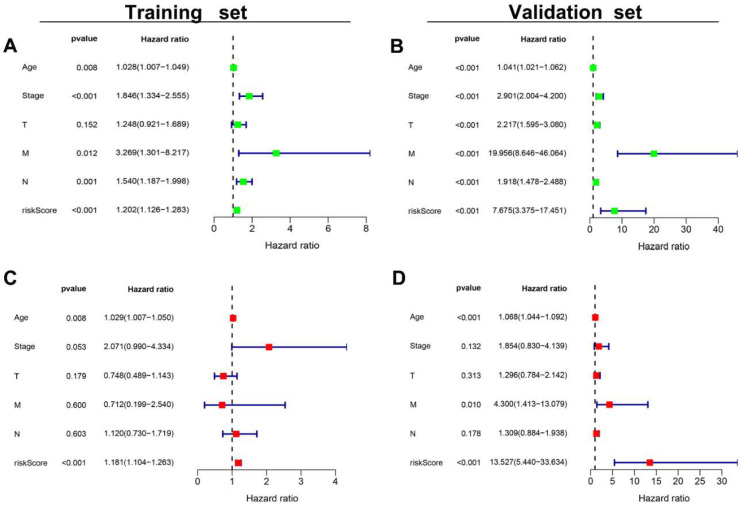
Evaluation of AR-LncM features. Univariate and multivariate Cox forest plots for risk scores and clinical characteristics in the training set (**A**,**C**) and the validation set (**B**,**D**). (**E**–**M**) Survival analysis of high- and low-risk groups adjusted for the clinical characteristics, such as age, metastatic status, lymph node status, tumor stage, and tumor size, of BC patients.

**Figure 7 jpm-13-00513-f007:**
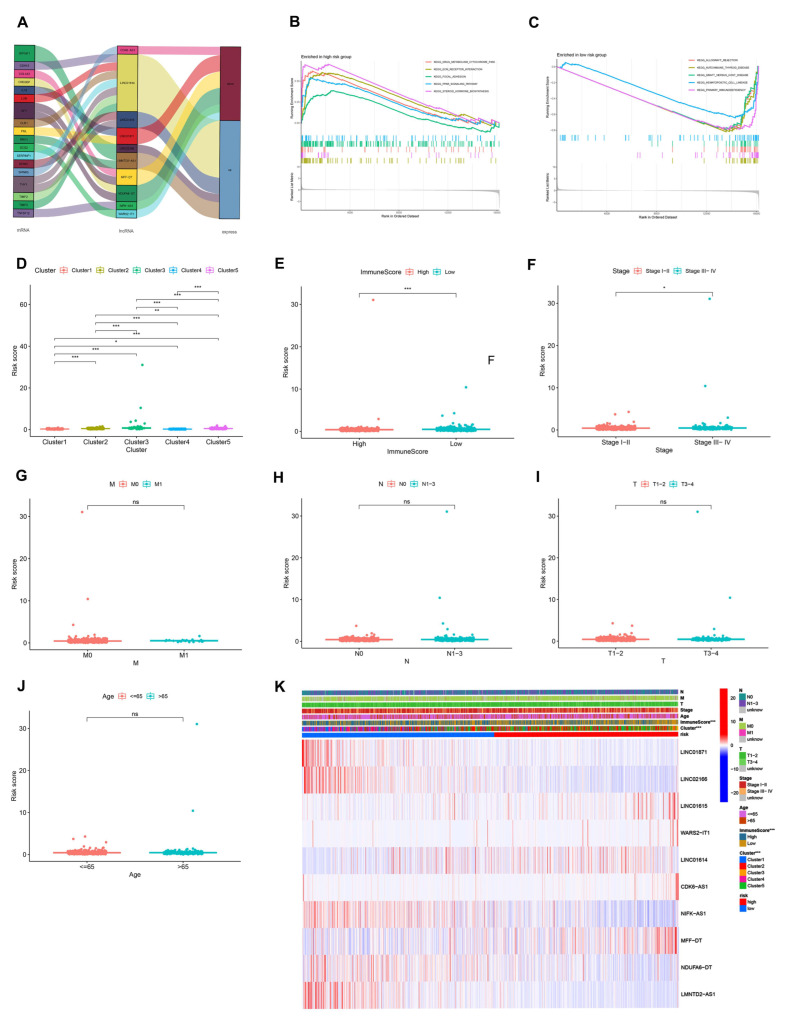
GSEA analysis and clinical evaluation of AR-LncM signatures. (**A**) Sankey diagram observed the correlation of lncRNAs in the model and their expression in cancer tissues and paracancerous tissues of BC patients. (**B**,**C**) Different pathways are enriched in low- and high-risk groups. (**D**–**J**) Differences in risk scores for different clinical features and different BC clusters. (**K**) Heatmaps reveal correlations between clinical features, BC clusters, ImmuneScore, and riskScore. ns, no significance; * *p* < 0.05, ** *p* < 0.01, and **** p* < 0.001.

**Figure 8 jpm-13-00513-f008:**
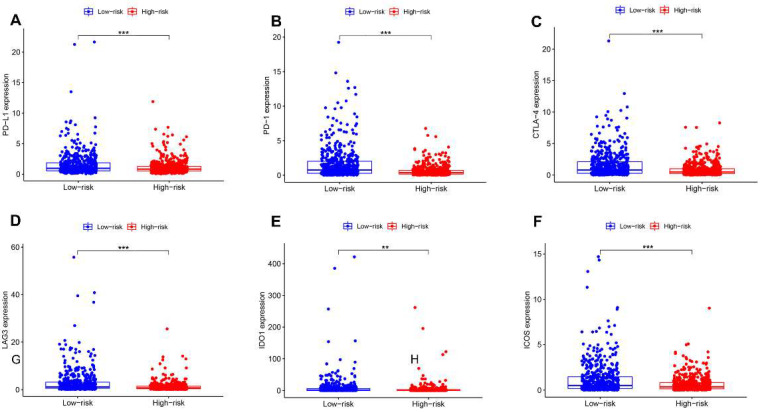
Comprehensive analysis of immune characteristics. (**A**–**F**) Expression of immune checkpoint members, including PD-L1, PD-1, CTLA-4, LAG3, IDO1, and ICOS, for the high- and low-risk BC groups. (**G**) M0 macrophages and M2 macrophages were elevated, while CD8 T cells, follicular helper T cells, regulatory T cells (Tregs), resting NK cells, and activated NK cells were decreased in the high-risk group compared with the low-risk group. (**H**–**L**) Different levels of immune cell infiltration in BC patients reveal differences in prognosis. (**M**) Immune-related effects in the high- and low-risk groups based on ssGSEA. (**N**–**P**) Violin plot of TIDE scores, TIDE exclusion score, and TIDE dysfunction scores. * *p* < 0.05, ** *p* < 0.01, and **** p* < 0.001.

**Figure 9 jpm-13-00513-f009:**
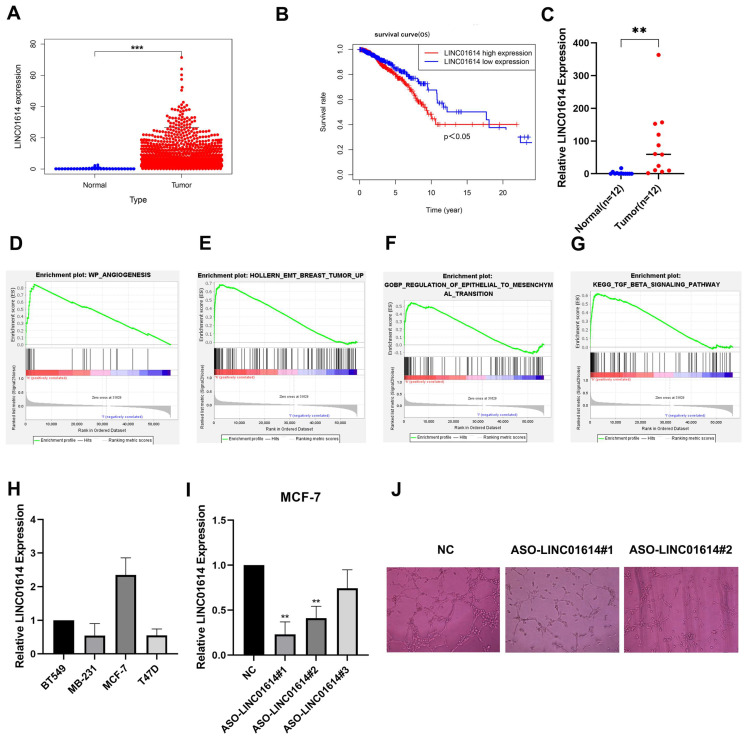
LINC01614 functional enrichment and in vitro experiments. (**A**) Expression of LINC01614 in the TCGA-BRCA dataset. (**B**) High expression of LINC01614 predicts poor prognosis in BC patients. (**C**) Expression of LINC01614 in cancerous and paracancerous tissues of BC patients. (**D**–**G**) Gene set enrichment analysis of LINC01614. (**H**) Expression of LINC01614 in breast cancer cell lines. (**I**) Knockdown of LINC01614 in MCF-7 cells. (**J**) Correlation between LINC01614 and angiogenesis. (**K**–**M**) Correlation of LINC01614 with invasion and metastasis of MCF-7 cells. (**N**) Western blotting revealed that LINC01614 regulates EMT. * *p* < 0.05, ** *p* < 0.01, and *** *p* < 0.001.

**Table 1 jpm-13-00513-t001:** Angiogenesis-related genes.

HALLMARK_ANGIOGENESIS	WP_ANGIOGENESIS	ANGIOGENESIS
APOH, APP, CCND2, COL3A1, COL5A2, CXCL6, FGFR1, FSTL1, ITGAV, JAG1, JAG2, KCNJ8, LPL, LRPAP1, LUM, MSX1, NRP1, OLR1, DGFA, PF4, PGLYRP1, POSTN, PRG2, PTK2, S100A4, SERPINA5, SLCO2A1, SPP1, STC1, THBD, TIMP1, TNFRSF21, VAV2, VCAN, VEGFA, VTN	AKT1, ANGPT1, ARNT, CREBBP, FGF2, FGFR2, FLT1, HIF1A, KDR, MAPK1, MAPK14, MMP9, NOS3, PDGFB, PDGFRA, PIK3CA, PLCG1, PTK2, SMAD1, SRC, TEK, TIMP2, TIMP3, EGFA	ACVRL1, AGGF1, AMOT, ANG, ANGPTL3, ANGPTL4, ATP5IF1, BTG1, C1GALT1, CANX, CDH13, CHRNA7, COL4A2, COL4A3, CXCL8, EGF, EMCN, EPGN, ERAP1, FOXO4, HTATIP2, IL17F, IL18, MYH9, NCL, NF1, NOTCH4, NPPB, NPR1, PF4, PLG, PML, PROK2, RHOB, RNH1, ROBO4, RUNX1, SCG2, SERPINF1, SHH, SPHK1, SPINK5, STAB1, TGFB2, THY1, TNFSF12, TNNI3, VEGFA

**Table 2 jpm-13-00513-t002:** Clinicopathological Characteristics of BC patients.

Cas NO.	Sex	Age, Years	Diagnosis	Tumor Location	Tumor Size (cm)	TNM Stage
1	Female	69	IBC	Left	3.0 × 3.0	T2N0M0
2	Female	49	IBC	Right	3.2 × 1.7	T2N1M0
3	Female	61	IBC	Right	1.9 × 1.7	T1N1M0
4	Female	55	IBC	Right	2.5 × 2.1	T2N0M0
5	Female	46	IBC	Right	2.0 × 1.0	T1N1M0
6	Female	48	IBC	Left	9.0 × 6.0	T3N2M1
7	Female	41	IBC	Left	2.0 × 1.0	T2NOM0
8	Female	38	IBC	Right	2.6 × 0.5	T2N1M0
9	Female	63	IBC	Left	2.1 × 1.6	T2NOM0
10	Female	61	IBC	Left	2.4 × 1.4	T2NOM0
11	Female	64	IBC	Right	2.9 × 1.7	T2NOM0
12	Female	55	IBC	Left	1.9 × 1.2	T1N1M0

IBC: invasive breast cancer.

**Table 3 jpm-13-00513-t003:** Details of 10 AR-lncRNAs used to construct prognostic signatures.

GENE	HR	*p* Value	Coeffient
LINC01871	0.805696513	0.000786757	−0.139091012
LINC02166	0.485364461	0.005052064	−0.504430065
LINC01615	1.131750977	0.009372695	0.046593523
WARS2-IT1	1.170943297	0.00455321	0.180717203
LINC01614	1.021172679	0.006761123	−0.005591609
CDK6-AS1	1.346135832	0.001023502	0.359540355
NIFK-AS1	0.664026974	0.005039487	−0.231976683
MFF-DT	8.040952442	0.000357726	1.310004855
NDUFA6-DT	0.212778995	0.004502171	−0.67551201
LMNTD2-AS1	0.889106279	0.008835066	−0.05439316

Abbreviation: HR, hazard ratios.

## Data Availability

Not applicable.

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
