# Peer review of "An Angiogenesis-Related lncRNA Signature Is Associated with Prognosis and Tumor Immune Microenvironment in Breast Cancer"

_jpm, 2023, doi:10.3390/jpm13030513_

Round 1
Reviewer 1 Report
Comments:
In Table 1 and 3, is necessary a footnote to the tables.
In Table 2, the title says “Clinicopathological Characteristics…”, there is an extra space between the two words.
In Table 2, in the last column it says “TMN stage”, should be “TNM stage”. In the same way, in the same column, but in line 5 it says “T1M1N0, keep the same TNM order of all the other results.
In the discussion, in general, it is necessary to place more emphasis on the data found and contrast with existing data already reported. The discussion is very short compared to all the results obtained, I suggest a more detailed discussion, where each of the results obtained are discussed.
Author Response
In Table 1 and 3, is necessary a footnote to the tables.
RESPONSE: Thank you very much for your valuable suggestions, we have added footnotes to the table and adjusted the table appropriately.
In Table 2, the title says “Clinicopathological Characteristics…”, there is an extra space between the two words.
RESPONSE: Thank you for your kind reminder, we have made corrections in Table 2
In Table 2, in the last column it says “TMN stage”, should be “TNM stage”. In the same way, in the same column, but in line 5 it says “T1M1N0, keep the same TNM order of all the other results.
RESPONSE: We apologize for our oversight, which we have corrected in Table 2
In the discussion, in general, it is necessary to place more emphasis on the data found and contrast with existing data already reported. The discussion is very short compared to all the results obtained, I suggest a more detailed discussion, where each of the results obtained are discussed.
RESPONSE: Thanks for your suggestions, we've improved our discussion section.

Reviewer 2 Report
This paper describes interesting work that could lead to some useful developments in the field. The description of the methodology used and the interpretation of some of the results need to be improved.
Abstract. It looks like the same methodology is described twice (lines 11-14 and 17-20). Can the description be condensed or, if they really are two different procedures, add text to make the difference clear?
Line 24. Based on the results in Figure 5, LINC01614 does not appear to be a representative gene in the model (see comment on Figure 5 Panel C).
Line 71. Rather than “Obtaining RNA-seq expression information and clinical information of BC patients”, the text should read “RNA-seq expression and clinical information was obtained”.
Lines 78-79. It is not clear what two variables were used to calculate the Spearman rank correlation.
Lines 79-80. This section should state what clinical endpoint was used in the Cox regression. It should also state what criteria derived from the Cox model were used to make the selection of AR-lncRNA’s. Did this selection lead to the 17 AR-lncRNA’s mentioned in line 96?
Line 100. What were the sizes of the two risk groups in the training set? The methods text should clarify if the models developed in the training set were applied without further modification to the test set. The text should also clarify what was done to get the results on the whole TCGA data set: was the model developed on the training set used or was a model fit to the entire data set?
Line 101. The text should specify what method was used to compute the ROC curves. There are several methods available for survival analysis, so it is important to specify which one was chosen.
Line 104. Which clinical characteristics were chosen for inclusion in the nomogram? How were they chosen?
Line 162. “Mean standard deviation” should read “mean and standard deviation”.
Line 163. “Statistical comparisons” here presumably refers specifically to comparisons using continuous numeric outcome variables. The text should make this clear.
Line 174. The hazard ratios in Figure 1E are hard to interpret because the reader doesn’t know the dynamic ranges of the AR-lncRNA’s. Expressing the results as standardized hazard ratios (proportional change in the hazard per one standard deviation of the AR-lncRNA expression) would make them interpretable.
Line 189. The difference in clinical prognosis between clusters 1 and 2 is not clear. What is the basis of the claim that cluster 1 has the best prognosis?
Line 191-192. The heat map shows no apparent differences in age, T-stage or N-stage among the clusters. What is the justification for this claim?
Lines 198-199. Although statistically significant, the magnitude of the correlation between PD-L1 and most of these lncRNA’s is apparently very small. What then is the basis of the claim that cluster 1 patients are more likely to respond to ICI therapy?
Figure 2L. A key to this figure should be provided explaining how to interpret the size of the color dots and what the presence of star means.
Line 209. The “ESTIMATE score” should be explained.
Lines 255-257. Statistical significance of the correlation cannot be used to conclude close relation, as even a correlation of very small magnitude can easily be statistically significant. It is the magnitude of the correlation that is important. Figure 5C examined the correlation between the risk score and each of its component lncRNAs. This gives a measure of the importance of each lncRNA in the risk score. From Figure 5C, it appears that CDK6-AS1 contributes by far the most to the risk score of all the LncRNA’s, followed by WARS2-IT1, LNC01615 and LMNTD2-AS1. LINC01614 appears to be extremely weakly correlated, suggesting that it contributes little to the risk score. Therefore, LINC01614 cannot be considered a “representative” lncRNA in the score (see comment on line 440).
Figure 5. The meaning of “*”, “**” and “***” need to be explained.
Figure 5, panel D. What is “NMA”? Is “Risk” the risk score from the LASSO regression using the lncRNA’s? If so, it would be clearer to call this “lncRNA risk score” or something similar through the figure.
Lines 269-271. It is not clear from Figure 5 that the LncRNA risk score is the best predictor. Age has a higher area under the ROC curve.
Figure 6. The figure should indicate which panels represent results from univariable regression and which represent results from multivariable regression.
Figure 6 Panels A-D. The hazard ratios for all covariates are much stronger in the test (validation) set than they are in the training set. This is highly unusual. Are the training set and validation set graphs reversed?
It would help the reader to standardize terminology on either “test set” or “validation set” throughout the manuscript. In the current version, now both terms are used.
Lines 308-309. This should be made into a complete sentence. For example, it could say “We examined the association between . . . “ or “We looked into the association between . . . “.
Figure 8. The comparisons between groups display p-values in this figure, but in previous analogous figures, asterisks were used (*, **, ***). It would be better to use one convention throughout the manuscript.
Line 321. Readers may not be familiar with the TIDE algorithm. It should be briefly described here.
Line 323. Shouldn’t this say “the correlation between risk groups and potential effectiveness of ICI”?
Figure 9 Panel B. It would be better to put the p-value comparing the two survival curves as an inset in the graph rather than as part of the title.
Lines 351-352. It is not clear how GSEA could being used to examine the association between LNC01614 and these gene sets. Text should be added to explain how this was done.
Line 383. This is not a complete sentence. It could be made into one by changing “to investigate” to “we investigated”.
Line 399. “the preceding study” should be “this study”.
Line 429. It appears that LINC01614 was selected for further work since it was the only one of several lncRNAs that showed an apparent survival difference. Thus, in addition to the small sample size mentioned here, multiple testing may have lead to a false positive result. This should be acknowledged here.
Line 440. According to the results in Figure 5 Panel C, LINC01614 cannot be considered a “representative gene in the model”.
Author Response
Dear Reviewer,
Thank you for your letter and the comments concerning our manuscript entitled “An angiogenesis-related lncRNA signature is associated with prognosis and tumor immune microenvironment in breast cancer”. The comments are valuable and very helpful for revising and improving our paper, as well as our research. We have studied all the comments carefully and made corrections in accordance with your comments. Revised portions are marked in red in the manuscript. The responses to the yourcomments are as follows:
This paper describes interesting work that could lead to some useful developments in the field. The description of the methodology used and the interpretation of some of the results need to be improved.
Abstract. It looks like the same methodology is described twice (lines 11-14 and 17-20). Can the description be condensed or, if they really are two different procedures, add text to make the difference clear?
RESPONSE: Dear reviewer, thank you for your kind reminder, I think I need to explain that the abstract part (lines 11-14 and 17-20) in the manuscript is not a repetitive description, lines 11-14 are an overview of the research methods, the purpose is to introduce the methods used in this study, and 17-20lines is to explain the results obtained by the above research methods, and the purpose is to simply explain the results of this study.
Line 24. Based on the results in Figure 5, LINC01614 does not appear to be a representative gene in the model (see comment on Figure 5 Panel C).
RESPONSE: Thanks for your comment, in Figure 5C is shown the correlation of the genes in the model with the risk score, indeed, we see that LINC01614 is not highly correlated with the risk score, while CDK6-AS1 has the highest correlation, we choose CDK6 -AS1 seems to be more reasonable as a representative gene. And why we choose LINC01614 is mainly based on the following points.
- Through the analysis, we found that CDK6-AS1 was lowly expressed in breast cancer, whereas LINC01614 was highly expressed. The high expression of LINC01614 predicted poor prognosis in breast cancer patients, which was very consistent with the characteristics of tumor biomarkers, while LINC01615, LINC02166 and LMNTD2-AS1 did not have this feature, as revealed in Supplementary Figure 1A-H.
- In our previous research, we also tried to verify CDK6-AS1, but because its expression in breast cancer tissues and cells was too low, we could not get accurate data through PCR, so we focused our research on the expression genes that are higher and suggest a poorer prognosis, which is convenient for the development of research.
- In other reports, the function of LINC01614 has been partially validated, for example, in a study by Radhakrishnan et al[1], LINC01614 was shown to be an unfavorable prognostic marker for breast cancer, associated with HR+/HER2+ BC molecular isoforms and regulated by TGFβ and FAK signaling; similarly, in a study by Wang et al[2], it was shown that high expression of LINC01614 was positively associated with several genomes, including TGF-β1 response, CDH1 signaling and cell adhesion pathways. All of the above studies suggest that LINC01614 may be a potential marker for breast cancer, suggesting that LINC01614 deserves to be explored in depth.
In summary, we believe that LINC01614 is very representative and worthy of further investigation.
Line 71. Rather than “Obtaining RNA-seq expression information and clinical information of BC patients”, the text should read “RNA-seq expression and clinical information was obtained”.
RESPONSE: Thank you very much for your suggestion, we have corrected this in the manuscript.
Lines 78-79. It is not clear what two variables were used to calculate the Spearman rank correlation.
RESPONSE: Dear reviewer, thank you for the reminder, and we apologize for not describing it in detail here. In fact, we used Spearman correlation analysis to explore the correlation between angiogenesis-related lncRNAs and angiogenesis-related genes, and we have revised this in the manuscript and clearly indicated the variables and criteria used.
Lines 79-80. This section should state what clinical endpoint was used in the Cox regression. It should also state what criteria derived from the Cox model were used to make the selection of AR-lncRNA’s. Did this selection lead to the 17 AR-lncRNA’s mentioned in line 96?
RESPONSE: Thank you very much for the reminder that here we used the AR-lncRNAs associated with OS for the univariate Cox regression analysis, which we have corrected in the manuscript; as you mentioned, the results obtained by univariate regression were exactly those 17 AR-lncRNAs that were strongly associated with patient prognosis.
Line 100. What were the sizes of the two risk groups in the training set? The methods text should clarify if the models developed in the training set were applied without further modification to the test set. The text should also clarify what was done to get the results on the whole TCGA data set: was the model developed on the training set used or was a model fit to the entire data set?
RESPONSE: Thank you very much for your reminder, based on your suggestion we have validated our data again and found that the number of patients with complete clinical information (including survival time and survival status) in the data we obtained was 1078 (Supplementary Table 1), therefore the number of 1096 patients in Line 24 was wrong, we apologize for our oversight and we have made the revisions. As you mentioned, our model was constructed based on the entire dataset and was applied without modifications to the test set, and in addition we provided data for both risk groups in the training set (Supplementary Table 2), and the table clearly shows the number of patients in the high-risk group (270) and the low-risk group (269), which we have also clearly labeled in the manuscript.
Line 101. The text should specify what method was used to compute the ROC curves. There are several methods available for survival analysis, so it is important to specify which one was chosen.
RESPONSE: Thank you for your kind reminder. Here we choose the timeROC software package to draw the ROC curve. The advantage of timeROC is that it can use the additional information of the onset time of each individual, and can construct the ROC curve at multiple time points and compare factor's predictive power. This enables us to know how long our survival analysis is effective, and how to group continuous variables in this time range is the most appropriate, so we choose timeROC to draw the ROC curve,we have made this clear in the manuscript.
Line 104. Which clinical characteristics were chosen for inclusion in the nomogram? How were they chosen?
RESPONSE: The clinical information we used in constructing the nomogram included the metastasis of the patient's tumor, the patient's lymph node involvement, the size of the patient's tumor, the patient's age, and the patient's risk score, all of which were readily available and widely used in constructing the nomogram. Since the data we obtained had fewer patients with distant tumor metastases, the difference could not be reflected in the nomogram, and we kept this part of the results in order to maintain the accuracy of the data. If necessary, we can provide the nomogram without tumor metastasis status.
Line 162. “Mean standard deviation” should read “mean and standard deviation”.
RESPONSE: Thank you for the reminder, we have made corrections in the manuscript
Line 163. “Statistical comparisons” here presumably refers specifically to comparisons using continuous numeric outcome variables. The text should make this clear.
RESPONSE: Thank you for the reminder, we have made corrections in the manuscript.
Line 174. The hazard ratios in Figure 1E are hard to interpret because the reader doesn’t know the dynamic ranges of the AR-lncRNA’s. Expressing the results as standardized hazard ratios (proportional change in the hazard per one standard deviation of the AR-lncRNA expression) would make them interpretable.
RESPONSE: Thank you for your suggestion. The results shown in Figure 1E refer to the methods in other literatures, such as Figure 1B in a study on breast cancer by Zhang et al.[3]; Figure 3C in a study on hepatocellular carcinoma by Yang et al.[4]; and Figure 2A in a study on thyroid carcinoma published in your journal also adopted the same method[5]. So, we think the current results are comprehensible to readers.
Line 189. The difference in clinical prognosis between clusters 1 and 2 is not clear. What is the basis of the claim that cluster 1 has the best prognosis?
RESPONSE: As you pointed out, there is no obvious difference in the OS of patients between cluster 1 and cluster 2, but there is a significant difference between the two clusters and cluster 3, cluster 4, and cluster 5. Combined with the PFS of the patients, it can be seen that cluster 1 has a clear advantage, so cluster 1 is considered to be better prognostic.
Line 191-192. The heat map shows no apparent differences in age, T-stage or N-stage among the clusters. What is the justification for this claim?
RESPONSE: Dear reviewer, if what you mean here is Figure 2D, what we described in the manuscript is that patients' age, T stage, and N stage are different among different clusters, not no difference.
Lines 198-199. Although statistically significant, the magnitude of the correlation between PD-L1 and most of these lncRNA’s is apparently very small. What then is the basis of the claim that cluster 1 patients are more likely to respond to ICI therapy?
RESPONSE: Thank you for your attention, firstly, the 17 AR-lncRNAs associated with prognosis are not unique to cluster1, here we just show the correlation of these AR-lncRNAs with PD-L1, and it is clear that there is a correlation between the two; secondly, our Figure 2J-K results clearly show that in patients with cluster1 the immune checkpoint members (including PD-1, CTLA-4, Tim3, LAG3, ICOS and IDO1) with high expression, which implies that patients with cluster 1 have higher immunogenicity and may respond to ICI treatment.
Figure 2L. A key to this figure should be provided explaining how to interpret the size of the color dots and what the presence of star means.
RESPONSE: Thank you for your suggestion, we have added a note in the figure notes that red in the figure indicates a positive correlation and blue indicates a negative correlation, with an asterisk indicating statistical significance, and a darker color means that this significance is more pronounced.
Line 209. The “ESTIMATE score” should be explained.
RESPONSE: In the tumor microenvironment, immune cells and stromal cells are the two main types of non-tumor components and have been shown to be of great value for tumor diagnosis and prognostic assessment. The immune score and stromal score calculated based on the ESTIMATE algorithm contribute to the quantification of the immune and stromal components in tumors. In this algorithm, the immune and stromal scores are calculated by analyzing specific gene expression profiles of immune and stromal cells to predict the degree of infiltration of non-tumor cells, which is the tumor purity score (ESTIMATE score). In the present study, we can learn from Figure 3A-B that cluster 1 has a higher immune score and stromal score, which in turn presumes that cluster 1 has a higher tumor purity, and Figure 3C reveals exactly this result.
Lines 255-257. Statistical significance of the correlation cannot be used to conclude close relation, as even a correlation of very small magnitude can easily be statistically significant. It is the magnitude of the correlation that is important. Figure 5C examined the correlation between the risk score and each of its component lncRNAs. This gives a measure of the importance of each lncRNA in the risk score. From Figure 5C, it appears that CDK6-AS1 contributes by far the most to the risk score of all the LncRNA’s, followed by WARS2-IT1, LNC01615 and LMNTD2-AS1. LINC01614 appears to be extremely weakly correlated, suggesting that it contributes little to the risk score. Therefore, LINC01614 cannot be considered a “representative” lncRNA in the score (see comment on line 440).
RESPONSE: We very much agree with you, the reason why LINC01614 was chosen as the representative gene in this study we have already explained in the previous question.
Figure 5. The meaning of “*”, “**” and “***” need to be explained.
RESPONSE: Thank you for your reminder, it has been clearly stated in the legend section.
Figure 5, panel D. What is “NMA”? Is “Risk” the risk score from the LASSO regression using the lncRNA’s? If so, it would be clearer to call this “lncRNA risk score” or something similar through the figure.
RESPONSE: 1. The main reason for showing MNA here is that too few patients with distant tumor metastasis could be included in the construction of the nomogram, so the item of distant metastasis in the figure is shown as NA, indicating that distant metastasis is not applicable to the construction of the nomogram, but to maintain the accuracy of the data, we kept this feature in the nomogram, and if necessary, we can provide the nomogram without the item of distant metastasis. 2. RISK Score has been modified in the manuscript and replaced it with AR-lncM risk score.
Lines 269-271. It is not clear from Figure 5 that the LncRNA risk score is the best predictor. Age has a higher area under the ROC curve.
RESPONSE: As you mentioned, the AUC value for patient age is 0.845, while the AUC value for the AR-lncM risk score is 0.785. It is clear that the AR-lncM risk score is not the best predictor, but it is a widely accepted fact that patient age is a more prominent risk factor in many tumors, and our results demonstrate exactly this feature.In addition, our main purpose of constructing AR-lncM here is to provide another possible strategy for predicting survival prognosis in breast cancer, which may not be the best, but this one may be effective.The AUC of the AR-lncM risk score in Figure 5F was 0.785, clearly demonstrating the good predictive performance of AR-lncM.Further, by splitting the TCGA dataset into training and validation sets and testing the predictive efficacy of AR-lncM in both, we also see that AR-lncM has good predictive performance
Figure 6. The figure should indicate which panels represent results from univariable regression and which represent results from multivariable regression.
RESPONSE: Thank you for your suggestion, we have clearly marked in the figure notes of Figure 6A-D
Figure 6 Panels A-D. The hazard ratios for all covariates are much stronger in the test (validation) set than they are in the training set. This is highly unusual. Are the training set and validation set graphs reversed?
RESPONSE: Thank you for your question. We have checked our data again and confirmed that the training and validation sets are not reversed and that there is indeed a stronger risk ratio in the training set than in the validation set. First of all, by reviewing the literature, I found a study showing that the validation set has a better risk ratio than the training set (DOI:10.7150/ijbs.45050.). In addition, we believe this is because both our training and validation sets are from the entire TCGA dataset, both obtained randomly 1:1, but if the TCGA dataset is unevenly sliced, or the training and test sets are unevenly distributed, this may lead to a risk ratio that may be stronger or weaker for the training set than the validation set, but still reflects the fact that AR-LncM can be used as an independent risk factor. This also indicates that our AR-LncM has good predictive function.
It would help the reader to standardize terminology on either “test set” or “validation set” throughout the manuscript. In the current version, now both terms are used.
RESPONSE: We noticed this issue and we have corrected it in the manuscript.
Lines 308-309. This should be made into a complete sentence. For example, it could say “We examined the association between . . . “ or “We looked into the association between . . . “.
RESPONSE: Thank you for your suggestion, we have modified this part in the manuscript.
Figure 8. The comparisons between groups display p-values in this figure, but in previous analogous figures, asterisks were used (*, **, ***). It would be better to use one convention throughout the manuscript.
RESPONSE: Thanks to your kind reminder, we have re-corrected Figure 8A-F and re-added it to the manuscript.
Line 321. Readers may not be familiar with the TIDE algorithm. It should be briefly described here
RESPONSE: Thank you for your suggestion, which we have explained in the methods section.
Line 323. Shouldn’t this say “the correlation between risk groups and potential effectiveness of ICI”?
RESPONSE: Thank you very much, your suggestion was concise and clear, and we have made corrections in the manuscript.
Figure 9 Panel B. It would be better to put the p-value comparing the two survival curves as an inset in the graph rather than as part of the title.
RESPONSE: Based on your suggestion, we have made corrections in Figure 9B.
Lines 351-352. It is not clear how GSEA could being used to examine the association between LNC01614 and these gene sets. Text should be added to explain how this was done.
RESPONSE: Thanks for your suggestion, according to your suggestion, we have made a correction in method 2.5.
Line 383. This is not a complete sentence. It could be made into one by changing “to investigate” to “we investigated”.
RESPONSE: Based on your suggestions, we have made changes in the manuscript.
Line 399. “the preceding study” should be “this study”.
RESPONSE: Based on your suggestions, we have made changes in the manuscript.
Line 429. It appears that LINC01614 was selected for further work since it was the only one of several lncRNAs that showed an apparent survival difference. Thus, in addition to the small sample size mentioned here, multiple testing may have lead to a false positive result. This should be acknowledged here.
RESPONSE: Thanks for your suggestion, we have added this statement to the manuscript.
Line 440. According to the results in Figure 5 Panel C, LINC01614 cannot be considered a “representative gene in the model”.
RESPONSE: We very much agree with you, the reason why LINC01614 was chosen as the representative gene in this study we have already explained in the previous question.

Round 2
Reviewer 1 Report
I do not have other comments.
Author Response
Dear reviewer, thank you for your recognition of our work, and wish you a happy life!
Reviewer 2 Report
COMMENTS ON REVISED MANUSCRIPT
Line 23. The arguments in your response for focusing on LINC01614 do not justify the term “representative”. A representative gene would somehow closely mirror the behavior of the model as a whole. The fact that this gene apparently has the strongest association with outcome does not make it representative. It would be more accurate to say “One gene in the model, LINC01614, . . . “.
Line 151. Your response states “our model was constructed based on the entire dataset and was applied without modifications to the test set, and in addition we provided data for both risk groups in the training set”. This procedure does not in any way validate the model. Instead, the model should be constructed based entirely on the training set without looking at the test set. This model should then be evaluated without modification on the test set. The performance of the model (hazard ratios for high vs. low risk, ROC’s, etc.) should be reported based on this procedure. This point of this procedure is to help eliminate bias that is caused by developing a model and then evaluating it on the same data. After doing this, it is acceptable to then refit the model based on the entire data set and report the coefficients for use in future research. However, assessing the performance characteristics of the refit model on these same data is biased and should be avoided.
Line 157. The revised manuscript does not contain any reference to the “timeROC” software mentioned in your response. The user documentation for “timeROC” should specify the statistical method used to calculated the ROC curve. The manuscript should state what that method is.
Line 270-271. The statement “AR-lncRNA clusters were closely associated with the patient’s age, T stage, and N stage (Figure 2D)” has no basis of support in Figure 2D, as no difference in these variables between clusters is notable. This statement should be removed from the manuscript.
Line 277-278. From the response to my question, it sounds like the basis of this statement is actually Figure 2E-K, not Figure 2L. These lines should be rewritten to make that clear.
Line 300. The ESTIMATE score should be explained in the manuscript so that the reader can understand what was done. Add text to this part of the manuscript, including references as appropriate, to explain the ESTIMATE score.
Figure 5D. The considerations of metastasis in the creation of the nomogram and why it was not included should be added to the text of the manuscript. It would be clearer to remove “MNA” from the figure, as this has no meaning to the reader.
Line 383. Given the explanation in your response, this sentence should say that “These findings suggest that AR-LncM is prognostic for the survival of IBC patients.” “Better” doesn’t make sense, since there is no comparison being done.
Figure 6. This and your prior explanations make it clear that the training and test sets were not used correctly. Please see my comment on Line 151 for an explanation on how they should be used.
Line 697. “LINC01614, a representative gene in the model” should simply say “LINC01614, a representative gene in the model”. “Representative” is not meaningful here.
Author Response
Line 23. The arguments in your response for focusing on LINC01614 do not justify the term “representative”. A representative gene would somehow closely mirror the behavior of the model as a whole. The fact that this gene apparently has the strongest association with outcome does not make it representative. It would be more accurate to say “One gene in the model, LINC01614, . . . “.
Response:Dear reviewer, thank you for your suggestion, we revised the most representative gene to the most interesting gene in the manuscript.
Line 151. Your response states “our model was constructed based on the entire dataset and was applied without modifications to the test set, and in addition we provided data for both risk groups in the training set”. This procedure does not in any way validate the model. Instead, the model should be constructed based entirely on the training set without looking at the test set. This model should then be evaluated without modification on the test set. The performance of the model (hazard ratios for high vs. low risk, ROC’s, etc.) should be reported based on this procedure. This point of this procedure is to help eliminate bias that is caused by developing a model and then evaluating it on the same data. After doing this, it is acceptable to then refit the model based on the entire data set and report the coefficients for use in future research. However, assessing the performance characteristics of the refit model on these same data is biased and should be avoided.
Response:Dear Reviewer, thank you for your kind reminder, it has been very rewarding for us to read your suggestions. We have checked our code and found that we did build the model based on the training set and our previous response was wrong. This problem occurred because we previously only checked the input file (expression matrix of 17 AR-lncRNAs) without carefully checking our code, which was shown to have 1078 samples in that input file, so we incorrectly thought we were building the model based on the whole dataset, and after carefully checking the code, we confirmed that we first randomly assigned the total dataset 1:1 (assigned as training set and validation set), and then built the model based on the training set, and finally validated it in the validation set. We sincerely apologize for our carelessness and for wasting your precious time.
Line 157. The revised manuscript does not contain any reference to the “timeROC” software mentioned in your response. The user documentation for “timeROC” should specify the statistical method used to calculated the ROC curve. The manuscript should state what that method is.
Response:Thanks for your suggestion, we have added references about timeROC to the manuscript, readers can get the detailed calculation method of timeROC from the references.
Line 270-271. The statement “AR-lncRNA clusters were closely associated with the patient’s age, T stage, and N stage (Figure 2D)” has no basis of support in Figure 2D, as no difference in these variables between clusters is notable. This statement should be removed from the manuscript.
Response:Thanks for your suggestion, we have removed this sentence in the manuscript.
Line 277-278. From the response to my question, it sounds like the basis of this statement is actually Figure 2E-K, not Figure 2L. These lines should be rewritten to make that clear.
Response:Thanks for the suggestion, we have made this clear in the manuscript.
Line 300. The ESTIMATE score should be explained in the manuscript so that the reader can understand what was done. Add text to this part of the manuscript, including references as appropriate, to explain the ESTIMATE score.
Response:Thanks for your suggestion, we have added this section to the manuscript.
Figure 5D. The considerations of metastasis in the creation of the nomogram and why it was not included should be added to the text of the manuscript. It would be clearer to remove “MNA” from the figure, as this has no meaning to the reader.
Response:Thanks for your suggestion, we have made modifications in Methods and Figure 5D.
Line 383. Given the explanation in your response, this sentence should say that “These findings suggest that AR-LncM is prognostic for the survival of IBC patients.” “Better” doesn’t make sense, since there is no comparison being done.
Response:Thank you for your suggestion, we have made corrections in the manuscript.
Figure 6. This and your prior explanations make it clear that the training and test sets were not used correctly. Please see my comment on Line 151 for an explanation on how they should be used.
Response:Thank you, we have learned a lot from your suggestions, which we will work on in our follow-up research.
Line 697. “LINC01614, a representative gene in the model” should simply say “LINC01614, a representative gene in the model”. “Representative” is not meaningful here.
Response:Following your earlier suggestion, we have replaced "representative" with "interesting" in the manuscript.
